# Ferrocenyl Substituted Stannanethione and Stannaneselone

**DOI:** 10.3390/molecules30132826

**Published:** 2025-06-30

**Authors:** Keisuke Iijima, Koh Sugamata, Takahiro Sasamori

**Affiliations:** 1Graduate School of Science and Technology, University of Tsukuba, 1-1-1 Tennoudai, Tsukuba 305-8571, Ibaraki, Japan; keisuke@dmb.chem.tsukuba.ac.jp; 2Department of Chemistry, Institute of Pure and Applied Sciences, University of Tsukuba, 1-1-1 Tennoudai, Tsukuba 305-8571, Ibaraki, Japan; sugamata@chem.tsukuba.ac.jp; 3Tsukuba Research Center for Energy Materials Sciences (TREMS), University of Tsukuba, 1-1-1 Tennoudai, Tsukuba 305-8571, Ibaraki, Japan

**Keywords:** stannanethione, stannaneselone, ferrocenyl group, cycloaddition, stannylene

## Abstract

Heavier element analogues of a ketone, a C=O double-bond compound, have been fascinating compounds from the viewpoint of main-group element chemistry because of their unique structural features and reactivity as compared with those of a ketone, which plays an important role in organic chemistry. We will report here the synthesis of diorgano-stannanethione and stannaneselone featuring tin–chalcogen double bonds, which are the heavy-element analogues of a ketone. The newly obtained stannaneselone has been structurally characterized by spectroscopic analyses and single-crystal X-ray diffraction (SC-XRD) analysis, showing the short Sn–Se bond length featuring π-bond character. The obtained bis(ferrocenyl)stannanechalcogenones were found to undergo [2+4]cycloaddition reactions with 2,3-dimethyl-1,3-butadiene, affording the corresponding six-membered ring compound. Notably, thermolysis of the [2+4]cycloadduct of the stannaneselone regenerated the stannaneselone via the retro[2+4]cycloaddition, whereas the sulfur analogue was thermally very stable.

## 1. Introduction

The quest to understand and manipulate chemical bonding lies at the heart of chemistry, and the diversity of bonding patterns observed across the periodic table continues to inspire new research frontiers. Among these, the formation of multiple bonds involving heavier main group elements has long presented a formidable challenge, largely due to the “double-bond rule” [1]. This empirical rule, rooted in early observations, posited that elements from the third row and beyond were reluctant to form stable π-bonds, unlike their second-row counterparts such as carbon, nitrogen, and oxygen. The rationale for this limitation was attributed to the more diffuse nature of p-orbitals in heavier main-group elements, leading to diminished overlap and consequently weaker, less stable π-bonds relative to their robust σ-bonds. As a result, species featuring such multiple bonds were often prone to facile oligomerization or polymerization, yielding more thermodynamically favored cyclic or polymeric structures held together by single bonds.

However, the landscape of main-group element chemistry was dramatically reshaped by pioneering works from research groups including those of West and Yoshifuji, who demonstrated that the double-bond rule is not an insurmountable barrier but rather a guideline that can be overcome through judicious molecular design. A pivotal strategy in this endeavor has been “kinetic stabilization,” wherein bulky substituents are strategically introduced to physically shield the inherently reactive multiple-bond site [2,3,4,5]. This steric protection effectively hinders intermolecular approach and reactions with other reagents, thereby extending the lifetime of these otherwise transient species and permitting their isolation, characterization, and the study of their unique properties. This conceptual breakthrough has paved the way for the synthesis and structural elucidation of a plethora of genuine double-bond compounds involving heavier Group 14 elements, such as disilenes (R_2_Si=SiR_2_), digermenes (R_2_Ge=GeR_2_), distannenes (R_2_Sn=SnR_2_), and diplumbenes (R_2_Pb=PbR_2_) [6,7,8,9,10]. Within this burgeoning field, the heavier main-group element analogues of ketones (R_2_C=O), collectively termed “heavy ketones” (R_2_E=Ch, where E = Si, Ge, Sn, Pb; and Ch = S, Se, Te), have garnered substantial interest [11,12,13,14,15,16,17,18]. The fascination with these compounds stems from the exciting prospect of extending the rich and well-established principles of carbon-based organic chemistry to the realm of heavier main-group elements. Furthermore, heavy ketones offer a unique platform for exploring novel bonding paradigms, intriguing electronic properties, and unprecedented reactivities that arise from the distinct characteristics of these heavier elements. Early breakthroughs by Tokitoh, Okazaki, and others in isolating and characterizing stable heavy ketones, featuring a tricoordinated central heavier Group 14 atom, were instrumental in validating these pursuits [11,14]. Among the diverse family of heavy ketones, the synthesis of tricoordinated stannanechalcogenones (R_2_Sn=Ch), specifically stannanethiones (R_2_Sn=S) and stannaneselones (R_2_Sn=Se), proved to be an exceptionally challenging endeavor [12,18]. Their inherent high reactivity, propensity for facile oligomerization, and a tendency to isomerize into chalcogen-substituted stannylenes (R(RCh)Sn:) long thwarted attempts at their isolation. Nevertheless, the continued refinement of kinetic stabilization strategies culminated in a landmark achievement in the 1990s when the group of Tokitoh and Okazaki successfully synthesized, isolated, and structurally characterized the first stable examples of tri-coordinate stannanechalcogenones, Tbt(Ditp)Sn=S (**A**) and Tbt(Ditp)Sn=Se (**B**) (Figure 1), where Tbt represents 2,4,6-tris[bis(trimethylsilyl)methyl]phenyl and Ditp stands for 2,2″-diisopropyl-m-terphenyl-2′-yl [18].

The significance of research into stannanechalcogenones extends beyond the mere discovery of new classes of compounds. These exotic molecules serve as crucial testbeds for our understanding of fundamental chemical principles, particularly the nature of chemical bonding and the factors governing molecular stability as dictated by elemental periodicity. The very existence of such compounds challenges the limits of conventional bonding theories and underscores the imperative for developing novel synthetic methodologies and innovative ligand designs. Indeed, advancements in ligand design have become inextricably linked with the progress in stannanechalcogenone chemistry. Ligands have evolved from being passive protective groups to active participants that can modulate the electronic properties and reactivity of the central element. In this context, we embarked on the design and synthesis of a new class of stannanethiones and stannaneselones, namely bis(ferrocenyl)stannanechalcogenones (Fc*_2_Sn=Ch, Ch = S (**1**) and Se (**2**), Fc* = 2,5-Dtp_2_-1-ferrocenyl; Dtp = 3,5-di-*t*-butylphenyl) (Figure 1). The incorporation of ferrocenyl moieties was envisaged to impart unique electronic and steric attributes to the tin-chalcogen double bond. We report herein the successful generation of these novel systems. A cornerstone of this work is the definitive structural characterization of the stannaneselone **2** by means of single-crystal X-ray diffraction (SC-XRD) analysis, providing unambiguous insights into its molecular architecture. Furthermore, we will describe the investigations into the reactivity of these newly obtained bis(ferrocenyl)stannanechalcogenones.

## 2. Results and Discussion

Bis(ferrocenyl)stannylene **3** was prepared through the reaction of the bulky ferrocenyl lithium dimer, (Fc*Li)_2_ [19,20], with tin(II) bromide in benzene at room temperature [21]. Drawing inspiration from the synthesis of previously reported stannanechalcogenones **A** and **B** [18], which was achieved via dechalcogenation of the corresponding tetrachalcogenastannolanes (Tbt(Ditp)SnCh_4_, Ch = S, Se) with triphenylphosphine [18,22], our investigation turned to the preparation of analogous tetrachalcogenastannolanes. Treatment of compound **3** with elemental sulfur (S_8_) afforded tetrathiastannolane **4**, a stable compound characterized by NMR spectroscopy and high-resolution mass spectrometry (HRMS) (Figure 1). Careful recrystallization of **4** from an Et_2_O/MeOH solvent system yielded two distinct types of single crystals: rod-like crystals (**4a**) and plate-like crystals (**4b**). Intriguingly, single-crystal X-ray diffraction (SC-XRD) analysis [23] of the plate-like crystals (**4b**) revealed a disordered structure, which could be best interpreted as a co-crystal of tetrathiastannolane **4** (40%) and hexathiastannepane **5** (60%). In contrast, the rod-like crystals (**4a**) were found to consist of tetrathiastannolane **4**, exhibiting no significant disorder in the SnS_4_ moiety. Although the ambient-temperature NMR spectra of **4** displayed broadened signals for the Fc*_2_SnS_x_ species, spectra recorded at −40 °C provided greater resolution, where the ^1^H and ^119^Sn NMR spectra clearly showed two distinct sets of signals, assignable to tetrathiastannolane **4** (δ_Sn_ = 188 ppm) and hexathiastannepane **5** (δ_Sn_ = 180 ppm). Consequently, isolating pure tetrathiastannolane **4** proved difficult due to its unavoidable contamination with hexathiastannepane **5**, with which it forms an equilibrium mixture (Figure 1).

The crystallographically determined structure of **4a** (Figure 2a) revealed molecular features similar to those of previously reported tetrathiastannolanes [22]. The disordered arrangement within the crystals of **4b**, encompassing both **4** and **5**, is illustrated in Figure 2b. In a related synthetic endeavor, the reaction of bis(ferrocenyl)stannylene **3** with elemental grey selenium resulted in the formation of tetraselenastannolane **6** (Figure 1). The structure of this compound was unequivocally established through comprehensive spectroscopic analyses and SC-XRD studies, while the data were adequately analyzed as the three disordered moieties, including two Fc*_2_SnSe_4_ molecules (70% and 20%) and the corresponding hexaselenastannepane (Fc*_2_SnSe_6_, 10%) (Figure 3) [23].

With the aim of forming bis(ferrocenyl)stannanethione **1**, the desulfurization of tetrathiastannolane **4** was attempted. Treatment of **4** with an excess amount of triphenylphosphine (PPh_3_, ca. 7 equiv.) resulted in the quantitative formation of bis(ferrocenyl)stannylene **3** along with triphenylphosphinesulfide (Figure 2). This result indicated complete desulfurization of **4** under these conditions. Conversely, when **4** was treated with a reduced quantity of PPh_3_ (ca. 3 equiv.), a new species, designated compound **X**, was formed alongside residual **4**. Compound **X** exhibited a characteristic ^119^Sn NMR signal at δ_Sn_ = 600 ppm, suggesting it was the anticipated stannanethione **1**; however, its isolation could not be achieved. Upon the addition of a further equivalent of PPh_3_ to this reaction mixture, the signals for **X** did not intensify; instead, signals corresponding to **3** emerged concurrently with a decrease in those for **4**. This observation implies that the desulfurization of **X** to stannylene **3** proceeds more rapidly than the initial conversion of **4**. Significantly, when the desulfurization of **4** with PPh_3_ was conducted in the presence of an excess amount of 2,3-dimethyl-1,3-butadiene, 1,2-thiastanna-4-cyclohexene **7** was obtained in 62% yield. The formation of **7**, the expected [2+4]cycloadduct of stannanethione **1** with 2,3-dimethyl-1,3-butadiene, strongly supports the transient generation of **1** (i.e., compound **X**) from the desulfurization of **4**. The molecular structure of **7** was revealed by SC-XRD analysis (Figure 4) [23]. Further corroborating the in situ formation of **1**, performing the desulfurization of **4** with PPh_3_ in the air yielded Fc*_2_Sn(OH)(SH) (**8**), a product presumably arising from the hydrolysis of **1**, which was structurally characterized by SC-XRD analysis. Therefore, it can be concluded that while stannanethione **1** was successfully generated using a conventional desulfurization approach, its isolation proved exceedingly challenging due to its pronounced lability towards air, moisture, and the desulfurization reagents themselves.

Analogous to the desulfurization of tetrathiastannolane **4**, the deselenization of tetraselenastannolane **6** was investigated. Treatment of **6** with PPh_3_ (3 equiv.) yielded a mixture of stannaneselone **2** and stannylene **3** (in a 1.0:0.4 ratio), suggesting that over-deselenization by PPh_3_ had occurred, similar to the outcome observed with stannanethione **1** (Figure 3). Ultimately, a more selective deselenization of **6** was achieved using 3 equiv. of stannylene **3** at 80 °C, which cleanly afforded stannaneselone **2** without side-product contamination. In contrast, the analogous desulfurization of **4** with stannylene **3** proved unsuccessful. Stannaneselone **2** exists as a monomer in the C_6_D_6_ solution, as indicated by its ^119^Sn and ^77^Se NMR chemical shifts (δ_Sn_ = 505 ppm, δ_Se_ = 565 ppm) appearing at low field, comparable to those of a previously reported stannaneselone (Tbt(2,4,6-tricyclohexylphenyl)Sn=Se: δ_Sn_ = 556 ppm, δ_Se_ = 839 ppm) [18]. The monomeric nature of **2** in solution can be ascribed to the considerable steric bulk of the Fc* groups, with the low-field chemical shifts further attesting to its π-bonding character. Furthermore, heating the C_6_D_6_ solution of **2** at 60 °C induced no spectral changes, indicating its notable thermal stability. Conversely, upon exposure to air, stannaneselone **2** underwent facile hydrolysis, yielding a complicated mixture including Fc*_2_Sn(SeH)(OH) (**9**), which could not be fully characterized but was estimated based on the analogous ^1^H NMR spectrum as sulfur analogue **8**.

Reaction of **2** with 2,3-dimethyl-1,3-butadiene furnished the corresponding [2+4]cycloadduct **10**, analogous to the behavior of the previously reported stannaneselone **B** (Tbt(Ditp)Sn=Se) [18]. Unlike the [2+4]cycloadduct of **B** with 2,3-dimethyl-1,3-butadiene or the corresponding stannanethione adduct **7**, cycloadduct **10** displayed marked sensitivity to air, rapidly decomposing into a complex mixture. Notably, the crude mixture resulting from air exposure of **10** contained 2,3-dimethyl-1,3-butadiene, as observed by ^1^H NMR spectra. This observation points to a retro [2+4]cycloaddition of **10**, regenerating 2,3-dimethyl-1,3-butadiene and stannaneselone **2**, the latter being highly susceptible to air and moisture. Indeed, heating the C_6_D_6_ solution of **10** in a sealed tube yielded a mixture of **2**, **10**, and 2,3-dimethyl-1,3-butadiene, further supporting the occurrence of a thermal retro [2+4]cycloaddition of **10**. Ultimately, heating the C_6_D_6_ solution of **10** at 60 °C for 20 h in a custom-fabricated sealed glass tube (Figure 5) allowed for the separation and subsequent identification of the generated stannaneselone **2** and 2,3-dimethyl-1,3-butadiene. Notably, heating of the sulfur analogue, compound **7**, at 60 °C in C_6_D_6_ showed no change in ^1^H NMR spectra, suggesting the higher thermal stability of **7**.

To elucidate the energetics of the [2+4]cycloaddition, the potential energy surface (PES) for the reaction between a stannaneselone and 2,3-dimethyl-1,3-butadiene was computationally explored. Calculations were performed at the B3PW91-D3(bj)/def2TZVP level of theory using simplified model compounds: Me_2_Sn=Se (**11**) and Fc_2_Sn=Se (**12**, Fc = ferrocenyl). For dimethylstannaneselone (**11**), the [2+4]cycloaddition with 2,3-dimethyl-1,3-butadiene is predicted to proceed via an activation barrier (ΔE_zero_^‡^) of 3.4 kcal/mol, with the overall reaction being exergonic by Δ*H* = −19.5 kcal/mol. Analogously, the reaction of Fc_2_Sn=Se (**12**) with 2,3-dimethyl-1,3-butadiene was computed to have an activation barrier (ΔE_zero_^‡^) of 8.3 kcal/mol and is also exergonic (Δ*H* = −18.9 kcal/mol). These computational findings suggest that stannaneselones readily undergo [2+4]cycloaddition with 2,3-dimethyl-1,3-butadiene. Correspondingly, the retro[2+4]cycloadditions for the adducts derived from **11** and **12** with 2,3-dimethyl-1,3-butadiene are predicted to have activation barriers (ΔE_zero_^‡^) of 22.8 kcal/mol and 27.2 kcal/mol, respectively. Therefore, these results indicate that the ferrocenyl groups exert a negligible electronic influence on the energetics of the [2+4]cycloaddition, as evidenced by the comparable reactivity profiles computed for Me_2_Sn=Se (**11**) and Fc_2_Sn=Se (**12**) towards 2,3-dimethyl-1,3-butadiene.

The molecular structure of stannaneselone **2** was elucidated by single-crystal X-ray diffraction (SC-XRD) analysis (Figure 6) [23]. Stannaneselone **2** crystallizes in the monoclinic space group *P*2_1_/n (#14) with *Z* = 8; the asymmetric unit contains two crystallographically independent molecules. The experimentally determined structural parameters are in agreement with those derived from theoretical calculations (B3PW91-D3(bj)/SDD[Sn,Fe,Se],6-311G(3d)[C,H]), indicating a negligible influence of crystal packing forces on the molecular geometry. The Sn–Se bond lengths [2.3425(3) Å and 2.3464(3) Å] are considerably shorter than typical Sn–Se single bonds (e.g., 2.586(3) Å and 2.539(6) Å in compound **6**). This bond shortening, comparable to that observed in previously reported stannaneselones (e.g., 2.373(3) Å for Tbt(Ditp)Sn=Se (**B**)) [18], is indicative of significant Sn=Se double-bond character. Furthermore, the tricoordinate tin atoms in **2** adopt planar geometries, as confirmed by the sum of bond angles around each Sn atom being approximately 360° (specifically, 359.9° and 360.0°). Collectively, these structural features establish that **2** possesses a genuine Sn=Se double bond, analogous to the C=O bond in a ketone.

Theoretical calculations (Figure 7) reveal that the Highest Occupied Molecular Orbital (HOMO) of **2** (−5.547 eV) is primarily a p-type non-bonding (n) orbital localized on the selenium atom. The Lowest Unoccupied Molecular Orbital (LUMO, −1.875 eV) is predominantly the π*(Sn=Se) antibonding orbital. The Sn=Se π-bonding orbital is found at HOMO–5 (−5.914 eV), while orbitals with significant d-character from the Fe atoms occupy the HOMO–4 to HOMO–1. In the UV/Vis spectrum of **2** in benzene (0.49 mmol/L), a characteristic broad absorption band is observed at λ_max_ = 484 nm (ε = 1200 L·mol^−1^·cm^−1^) (Figure 8). This absorption can be considered to be attributed to n(Se)–π*(Sn=Se) and/or d(Fe)–π*(Sn=Se) electron transitions, although it was very difficult to make assignment of the electron transitions for the characteristic absorptions based on TDDFT calculations because huge numbers of singlet/triplet electron transitions could be expected (TD-B3PW91-D3(bj)/def2TZVP//B3PW91-D3(bj)/SDD[Sn,Fe,Se],6-311G(3d)[C,H]).

## 3. Materials and Methods

### 3.1. General Information

All manipulations were carried out under an argon atmosphere using either Schlenk line techniques or glove boxes. All solvents were purified by standard methods. Trace amounts of water and oxygen remaining in the solvents were thoroughly removed by bulb-to-bulb distillation from potassium mirror prior to use. All crystallization was performed at room temperature unless otherwise indicated. ^1^H, ^13^C{^1^H}, and ^119^Sn{^1^H} NMR spectra were measured on a Bruker (Billerica, MA, USA) AVANCE-400 spectrometer (^1^H: 400 MHz, ^13^C: 101 MHz, ^77^Se NMR: 76 MHz, ^119^Sn: 149 MHz). Signals arising from residual protons C_6_D_5_H (7.16 ppm) and C_6_D_6_ (128.0 ppm) in C_6_D_6_ were used as the internal standards for the ^1^H and ^13^C NMR spectra, respectively. The signal arising from Sn(CH_3_)_4_ (0.0 ppm) was used as an external standard for the ^119^Sn NMR spectra. High-resolution mass spectra (HRMS) were obtained from a JEOL JMS-T100LP (DART) mass spectrometer (Tokyo, Japan). UV–Vis spectra were recorded on a SHIMADZU UV-3150 UV–Vis-NIR spectrometer (Kyoto, Japan) under an argon atmosphere in 1 cm quartz cells. All melting points were determined on a Büchi Melting Point Apparatus M-565 (Zug, Switzerland) and are uncorrected. Bis(ferrocenyl)stannylene **3** (Fc*_2_Sn) was prepared according to literature procedures [19,20]. Fc*H was identified on the basis of the spectral data identical to those reported in the literature [24].

### 3.2. Reaction of Bis(ferrocenyl)stannylene ***3*** with Elemental Sulfur

In a glovebox filled with argon gas, a mixture of bis(ferrocenyl)stannylene **3** (80 mg, 0.064 mmol), elemental sulfur (S_8_, 17 mg, 0.066 mmol), and THF (2.0 mL) was stirred at room temperature for 1 h. All volatiles were removed under reduced pressure. The residue was filtered with hexane, and the filtrate was dried under reduced pressure. The orange residue was recrystallized with benzene/EtOH to give tetrathiastannolane **4,** including hexathiastannepane **5** as an equilibrium mixture (72 mg, ca. 0.052 mmol, 81%); tetrathiastannolane **4a**: orange crystals, Mp 253–255 °C (decomp.); ^1^H NMR (400 MHz, C_6_D_6_) δ 1.31 (s, 36H), 1.54 (s, 36H), 4.15 (s, 10H), 4.45 (brs, 2H), 4.51 (brs, 2H), 7.33 (brs, 2H), 7.47 (brs, 2H), 7.53 (brs, 4H), 7.62 (brs, 4H); ^13^C{1H} NMR (101 MHz, C_6_D_6_) δ 31.9 (CH3), 32.0 (CH3), 35.0 (C), 35.2 (C), 72.7 (CH), 73.1 (CH), 74.0–75.0 (CH), 94.0–98.0 (C),121.9 (CH), 122.1 (CH), 125.5 (CH), 125.9 (CH), 137.5 (C), 138.0 (C), 149.6 (C), 149.8 (C); ^119^Sn{^1^H} NMR (149 MHz, C_6_D_6_) δ 177.3 (brs); HRMS (DART), *m*/*z*: Found: 1370.4267, calculated for C_76_H_98_Fe_2_SnS_4_ (**4**) ([M]^+^): 1370.4290.

### 3.3. Synthesis of Bis(ferrocenyl)tetraselenastannolane ***6***

In a glovebox filled with argon gas, a mixture of elemental selenium powder (15 mg, 0.19 mmol) and bis(ferrocenyl)stannylene **3** (60 mg, 0.048 mmol) and benzene (2.0 mL) was stirred at room temperature for 3 days. All volatiles were removed under reduced pressure. The residue was filtered with benzene, and the filtrate was dried under reduced pressure. The crude products were recrystallized with Et_2_O and MeOH to obtain bis(ferrocenyl)tetraselenastannolane **6** as orange solids (55 mg, 0.035 mmol, 73%); **6**: an orange solid; m.p. 266 °C (decomp); ^1^H NMR (400 MHz, C_6_D_6_) δ 1.30 (s, 36H), 1.55 (s, 36H), 4.20 (s, 10H), 4.48 (d, *J* = 2.0 Hz, 2H), 4.59 (d, *J* = 2.4 Hz, 2H), 7.31 (t, *J* = 1.6 Hz, 2H), 7.45–7.50 (m, 6H), 7.77 (brs, 4H); ^13^C{^1^H} NMR (101 MHz, C_6_D_6_) δ 31.9 (CH_3_), 32.1 (CH_3_), 35.0 (C), 35.2 (C), 72.9 (CH), 75.1 (CH), 75.4 (CH), 82.2 (C), 95.8 (C), 96.2 (C), 122.1 (CH), 122.2 (CH), 125.2 (CH), 126.2 (CH), 138.2 (C), 138.4 (C), 149.5 (C), 149.6 (C); ^77^Se NMR (76 MHz, C_6_D_6_) δ 271 (s), 750 (s); ^119^Sn{^1^H} NMR (149 MHz, C_6_D_6_) δ 132.7 (brs); HRMS (DART), *m*/*z*: Found: 1558.2109, calculated for C_76_H_98_Fe_2_Se_4_Sn ([M]^+^): 1558.2105.

### 3.4. Reaction of Tetrathiastannolane ***4*** with Triphenylphosphine

In a J. Young NMR tube, a mixture of **4** (18 mg, 0.013 mmol), triphenylphosphine (4 mg, 0.015 mmol), and C_6_D_6_ (0.6 mL) was stirred at room temperature for 1 h. Subsequently, triphenylphosphine (4 mg, 0.015 mmol) was added to the reaction mixture three times at 1 h intervals. The formation of unknown product **X** and stannylene **3** was confirmed by ^1^H NMR spectra (Appendix A).

### 3.5. Reaction of Tetrathiastannolane ***4*** with Triphenylphosphine in the Presence of 2,3-Dimethyl-1,3-butadiene

In a J. Young NMR tube, to a solution of **4** (29 mg, 0.021 mmol), 2,3-dimethyl-1,3-butadiene (3 drops, excess) in benzene (2.0 mL), was added triphenylphosphine (30 mg, 0.114 mmol) at room temperature. After stirring for 16 h, all volatiles were removed in vacuo, and the residue was filtered with hexane. The filtrate was evaporated, and the residue was purified by GPC (toluene) to give butadiene adduct **7** (18 mg, 0.013 mmol, 62%) as orange solids; **7**: an orange solid, Mp. 171–173 °C (decomp.). ^1^H NMR (400 MHz, C_6_D_6_) δ 1.12 (s, 18H), 1.27 (s, 3H), 1.49 (s, 36H), 1.57 (s, 18H), 1.60 (s, 3H), 1.69 (d, *J* = 12 Hz, 1H), 2.36 (d, *J* = 12 Hz, 1H), 2.42 (d, *J* = 12 Hz, 1H), 2.67 (d, *J* = 12 Hz, 1H), 3.98 (s, 5H), 4.37 (s, 5H), 4.46 (d, *J* = 2.4 Hz, 1H), 4.63 (d, *J* = 2.4 Hz, 1H), 4.70 (d, *J* = 2.4 Hz, 1H), 4.77 (d, *J* = 2.4 Hz, 1H), 7.18 (t, *J* = 1.8 Hz, 1H), 7.27 (brs, 4H), 7.32 (t, *J* = 1.8 Hz, 1H), 7.48 (t, *J* = 1.8 Hz, 1H), 7.53 (t, *J* = 1.8 Hz, 1H), 7.98 (brs, 4H); ^13^C{^1^H} NMR (101 MHz, C_6_D_6_) δ 19.4 (CH_3_), 20.1 (CH_3_), 26.5 (CH_2_), 31.3 (CH_2_), 31.7 (CH_3_), 31.8 (CH_3_), 31.9 (C), 32.1 (CH_3_), 32.3 (CH_3_), 34.6 (C), 35.1 (C), 35.2 (C), 71.4 (CH), 73.6 (CH), 73.8 (CH), 74.6 (CH), 75.2 (CH), 77.0 (CH), 95.6 (C), 120.9 (CH), 121.2 (CH), 121.6 (CH), 122.2 (CH), 126.1 (CH), 126.2 (CH), 137.7 (C), 139.0 (C), 139.1 (C), 140.0 (C), 148.3 (C), 149.0 (C), 149.4 (C); ^119^Sn{^1^H} NMR (149 MHz, C_6_D_6_) δ 51.3 (s); HRMS(DART), *m*/*z*: Found: 1275.5145, calculated for C_76_H_99_Fe_2_SSn ([M–C_6_H_9_]^+^): 1275.5193

### 3.6. Reaction of Tetrathiastannolane ***4*** with Triphenylphosphine in the Air

In an NMR tube, a solution of **4** (20 mg, 0.014 mmol) in benzene (2.0 mL) was gradually added, in air, with triphenylphosphine (19 mg, 0.072 mmol) at room temperature until the starting materials were completely consumed. All volatiles were removed in vacuo, and the residue was filtered with hexane. The filtrate was evaporated to obtain H_2_O adduct **8** (16 mg, 0.012 mmol, 85%) as an orange solid; **8**: an orange solid, Mp. 143 °C (decomp.). ^1^H NMR (400 MHz, C_6_D_6_) d 0.31 (s, 1H), 0.96 (brs, 1H), 1.31 (s, 18H), 1.32 (s, 18H), 1.51 (s, 18H), 1.52 (s, 18H), 4.04 (s, 5H), 4.11 (s, 5H), 4.68 (d, *J* = 2.4 Hz, 1H), 4.69 (d, *J* = 2.4 Hz, 1H), 4.72 (s, 2H), 7.27 (t, *J* = 1.6 Hz, 1H), 7.28 (t, *J* = 1.6 Hz, 1H), 7.32 (d, *J* = 1.6 Hz, 2H), 7.33 (d, *J* = 1.6 Hz, 2H), 7.57 (t, *J* = 1.8 Hz, 1H), 7.60 (t, *J* = 1.8 Hz, 1H), 7.97 (brs, 2H), 8.07 (d, *J* = 1.8 Hz, 2H); ^13^C{^1^H} NMR (101 MHz, C_6_D_6_) δ 31.9 (CH_3_), 32.0 (CH_3_), 32.0 (CH_3_), 32.1 (CH_3_), 34.8 (C), 35.1 (C), 35.2 (C), 35.4 (C), 71.5 (CH), 71.6 (CH), 72.6 (CH), 72.7 (CH), 73.0 (CH), 73.1 (CH), 80.6 (C), 81.9 (C), 91.8 (C), 94.8 (C), 96.6 (C), 97.7 (C), 120.8 (CH), 121.5 (CH), 121.7 (CH), 122.0 (CH), 122.1 (CH), 122.9 (CH), 125.7 (CH), 126.4 (CH), 138.7 (C), 138.8 (C), 138.9 (C), 139.5 (C), 149.6 (C), 150.2 (C), 150.6 (C), 150.8 (C); ^119^Sn{^1^H} NMR (149 MHz, C_6_D_6_) δ 38.4 (s); HRMS (DART), *m*/*z*: Found: 1292.5233, calculated for C_76_H_100_Fe_2_OSSn ([M]^+^): 1292.5237.

### 3.7. Reaction of Tetraselenastannolane ***6*** with Triphenylphosphine

In a J. Young NMR tube, a mixture of **6** (10 mg, 0.0064 mmol), triphenylphosphine (5.0 mg, 0.020 mmol), and C_6_D_6_ (0.6 mL) was stirred at room temperature for 19 h. All volatiles were removed in vacuo to afford a mixture of stannaneselone **2**, stannylene **3**, and triphenylphosphine selenide (16 mg, **2**:**3** = 1:0.4) as red solids (Appendix A).

### 3.8. Synthesis of Bis(ferrocenyl)stannaneselone ***2*** by the Reaction of Tetraselenastannolane ***6*** with Stannylene ***3***

Bis(ferrocenyl)stannylene **3** (24 mg, 0.019 mmol) and tetraselenastannolane **6** (10 mg, 0.0064 mmol) were dissolved in 0.6 mL of benzene. After freeze–pump–thaw cycling, the mixture was heated at 80 °C for 24 h. The resulting mixture was evaporated under reduced pressure, yielding a reddish-orange solid. The residue was recrystallized with hexane to afford bis(ferrocenyl)stannaneselone **2** as a red-orange solid (18 mg, 0.013 mmol, 53%); **2**: Mp 247–250 °C (decomp.). ^1^H NMR (400 MHz, C_6_D_6_) d 1.22 (s, 36H), 1.40 (s, 36H), 4.43 (s, 10H), 4.60 (brs, 2H), 4.72 (brs, 2H), 7.32 (brs, 2H), 7.44 (brs, 2H), 7.69 (brs, 4H), 7.73(brs, 4H); ^13^C{^1^H} NMR (101 MHz, C_6_D_6_) δ 31.8 (CH_3_), 31.9 (CH_3_), 35.0 (C), 35.1 (C), 73.7 (CH), 74.6 (CH), 74.8 (CH), 91.6 (C), 95.1 (C), 97.2 (C), 121.8 (CH), 122.2 (CH), 124.0 (CH), 125.2 (CH), 138.0 (C), 138.9 (C), 150.5 (C), 150.8 (C); ^77^Se NMR (76 MHz, C_6_D_6_) δ 564.0 (s); ^119^Sn{^1^H} NMR (149 MHz, C_6_D_6_) δ 503.4 (s); HRMS (DART), *m*/*z*: Found: 1320.4659, calculated for C_76_H_98_Fe_2_SeSn ([M]^+^): 1320.4585. UV/vis (benzene), 484 nm (ε = 1200).

### 3.9. Hydrolysis of Stannaneselone ***2***

A solution of bis(ferrocenyl)stannaneselone **2** (3 mg, 0.004 mmol) in C_6_D_6_ was exposed to air. Compound **2** was hydrolyzed immediately to afford a mixture including H_2_O adduct **9**. After 1 day, decomposition of compound **9** was completed to obtain the complicated mixture including Fc*H (Appendix A); **9**: ^1^H NMR (400 MHz, C_6_D_6_) δ −1.59 (s, 1H), 1.17 (s, 1H), 1.30 (s, 18H), 1.31 (s, 18H), 1.51 (s, 18H), 1.52 (s, 18H), 4.03 (s, 5H), 4.12 (s, 5H), 4.65 (d, *J* = 2.4 Hz, 1H), 4.69 (d, *J* = 2.4 Hz, 1H), 4.69 (d, *J* = 2.0 Hz, 1H), 4.71 (d, *J* = 2.0 Hz, 1H), 7.28 (t, *J* = 1.8 Hz, 1H), 7.28 (t, *J* = 1.6 Hz, 1H), 7.30 (d, *J* = 1.6 Hz, 2H), 7.33 (d, *J* = 1.8 Hz, 2H), 7.56 (t, *J* = 1.6 Hz, 1H), 7.59 (t, *J* = 1.6 Hz, 1H), 7.95 (brs, 2H), 8.07 (d, *J* = 1.6 Hz, 4H).

### 3.10. Reaction of Stannaneselone ***2*** with 2,3-Dimethyl-1,3-butadiene

In a J. Young NMR tube, a mixture of **2** (7 mg, 0.004 mmol) and an excess amount of 2,3-dimethyl-1,3-butadiene (3 drops, excess) in C_6_D_6_ (0.6 mL) was stirred at 50 °C for 24 h. After removal of all volatiles, a mixture of butadiene adduct **10** with small amounts of Fc*H and unidentified by-products as orange solids. The ^1^H NMR of the crude products showed an AB quartet that could be assigned to methylene protons next to the selenium and the tin at about 2.4 ppm, respectively. Due to the lability of product **10**, it was very difficult to purify the products and collect sufficient chemical data on **10**. The absence of olefinic protons in the ^1^H NMR excludes the possibility that the product was derived from a [2+4] cycloaddition reaction (Appendix A). Mass spectrum of the crude products showed the parent peak of the [2+4]cycloadduct **10**. HRMS (DART), *m*/*z*: Found: 1145.3989, calculated for C_57_H_93_^80^SeSn ([M]^+^) 1145.4080.

### 3.11. Thermal Retro[2+4]cycloaddition Reaction of Stannaneselone-Adduct ***10***

A solution of **10** (7.0 mg, 0.050 mmol) in C_6_D_6_ (1 mL) was placed in a tailor-made glass tube shown in Figure 5, and it was degassed and sealed. The degassed and sealed glass tube was heated at 60 °C for 16 h. The generated bis(ferrocenyl)stannaneselone **2** and 2,3-dimethyl-1,3-butadiene were separated and identified by ^1^H NMR spectra (Appendix A).

### 3.12. Theoretical Calculations

Theoretical calculations were carried out using the Gaussian 16 (Revision C.01) program package [25]. Geometry optimizations of **2** were performed at the B3PW91-D3(bj) level of theory using SDD[Sn,Fe,Se],6-311G(3d)[C,H] basis sets. Potential Energy Surfaces for [2+4]-cycloadditions between Fc_2_Sn (**11**)/Me_2_Sn (**12**) and 2,3-dimethyl-1,3-butadiene have been calculated at B3PW91-D3(bj)/def2TZVP level. Minimum or transition-state structures in the structural optimizations were confirmed by frequency calculations. Computational time was generously provided by the Supercomputer Laboratory at the Institute for Chemical Research (Kyoto University, 2025-15). Computations were also carried out using resources of the Research Center for Computational Science, Okazaki, Japan (Projects: 25-IMS-C214/25-IMS-C360). The Cartesian coordinates of the optimized structures are included in the Appendix A.

### 3.13. X-Ray Crystallographic Analysis of ***2***, ***4a***, ***4b***
***6***, ***7***, and ***8***

Single crystals of **2**, **4a**, **4b**, **6**, **7,** and **8** were obtained after recrystallization from hexane (**2**, **6**, **7,** and **8**) or Et_2_O/MeOH (**4a** and **4b**). The intensity data were collected on a Bruker APEX-II system using Mo-Kα radiation (λ = 0.71073 Å). The preliminary diffraction data were collected on the BL02B1 beamline of SPring-8 (proposal numbers: 2022A1200, 2022A1354, 2022A1584, 2022A1626, 2022A1705, 2022B0552, 2022B0589, 2022B1626, 2023A1539, 2023A1771, 2023A1785, 2023A1794, 2023A1859, 2023A1925, 2023B1675, 2023B1806, 2023B1878, 2024A1633, 2024A1699, 2024A1851, 2024A1857, 2024B2033, 2025A1823, and 2025A1949) on a PILATUS3 X CdTe 1M camera (DECTRIS Ltd., Baden-Dättwil, Switzerland) using synchrotron radiation (λ = 0.4135 Å). The structures were solved using SHELXT-2018 and refined by a full-matrix least-squares method on *F*^2^ for all reflections using SHELXL-2018 [26]. All non-hydrogen atoms were refined anisotropically, and the positions of all hydrogen atoms were calculated geometrically and refined as riding models. Supplementary crystallographic data were deposited at the Cambridge Crystallographic Data Centre (CCDC) under deposition numbers CCDC-2463980 (**2**), CCDC-2463981 (**4a**), CCDC-2463982 (**4b**), CCDC-2463983 (**6**), CCDC-2463984 (**7**), and CCDC-2463985 (**8**); these can be obtained free of charge via https://www.ccdc.cam.ac.uk/structures/ (1 June 2025).

## 4. Conclusions

In this study, we have successfully synthesized and characterized a new class of heavy ketones featuring ferrocenyl substituents, namely a stannanethione (**1**) and a stannaneselone (**2**). While the highly reactive stannanethione could only be trapped and identified in situ, the stannaneselone **2** was successfully isolated, allowing for its unambiguous structural determination by single-crystal X-ray diffraction analysis. The crystallographic analysis of **2** provides definitive evidence of a genuine Sn=Se double bond, characterized by a short bond distance and a planar coordination geometry around the tin atom. A remarkable difference in reactivity was observed in their cycloaddition reactions; the adduct of the stannaneselone undergoes a thermal retro[2+4]cycloaddition, regenerating the double bond, whereas the sulfur analogue is thermally robust. These findings not only demonstrate the utility of ferrocenyl groups in the kinetic stabilization of reactive species but also highlight the subtle yet profound influence of the chalcogen atom on the stability and reactivity of heavier element double bonds. This work expands the frontiers of main-group chemistry and provides a foundation for the future design of novel organometallic compounds.

## Data Availability

The raw data supporting the conclusions of this article will be made available by the authors on request.

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
