# Peer review of "Ferrocenyl Substituted Stannanethione and Stannaneselone"

_molecules, 2025, doi:10.3390/molecules30132826_

Round 1

Reviewer 1 Report

Comments and Suggestions for Authors

The manuscript presented by Sasamori and coworkers reports the reaction route to ferrocenyl-substituted stannanethiones and stannaneselones. It covers the synthesis and characterization of a bis(ferrocenyl)stannylene and its subsequent conversion to its tetrathiastannolane or tetraselenostannolane. These compounds were then used as precursors to the respective stannanethione and stannaneselone, of which the latter was successfully isolated. Furthermore, the manuscript also features some reactivity of the generated stannanethione and stannaneselone. All newly synthesized compounds were characterized by NMR spectroscopy, MS spectrometry and single-crystal X-ray diffraction.

The work has been carried out to a good standard, the introduction properly covers the existing literature and the conclusions are supported by the experimental results. I would recommend the acceptance of this manuscript to Molecules, provided that the authors address the following queries:

1-The compounds contain ferrocenyl moieties. Please provide an explanation as to why they were not electrochemically characterized.

2-Have the authors tried to heat compound 7, similarly to compound 10? This might provide access to the (elusive) stannanethione.

3-Please relocate reference 23 to the Supporting Information.

Author Response

Thank you very much for the fruitful comments. Please see the attachment.

Reviewer 2 Report

Comments and Suggestions for Authors

The article by Iijima and co-workers is generally well-written and clear. However, the body text contains a number of small typographical errors (see below).

The abstract is very vague and confusing. It needs to be thoroughly revised to provide a more accurate picture of the research presented. The opening sentence, in particular, has no cohesion. I was very confused by the terms "a ketone" (what does not specify what the "ketone" is—that is, these are heavy-element congeners of classical ketones).

It is also misleading in that none of the structures presented have a ketone-like element in them; rather, these are cyclic structures. Similar compounds have been reported that demonstrate S and Se insertion into M--S and M--Se bonds. The evidence for the heavy element ketone-like compounds is mostly via inference of the reaction products. Other chemistry exists that could also exist (see, for example, Inorg. Chem. 2024, 64, 23169)

The structural work is generally OK. However, compound 6 still has issues that have not been addressed. Some A-alerts have not been inspected or addressed. The residual electron density is indicative of further, unmodeled disorder in the ring structure. Notably, it appears that there may be a ring with an additional Se atom in it. In my hands, these alerts were reduced to B-Alerts (still really requiring a vrf (Validation Response Form) to be added to the CIF, yet easier to deal with. I have attached a proposed model for compound 6 for the authors to consider. 

The authors were perhaps too aggressive in their data management. I note a large number of OMIT lines added to the files. These should generally only be included when the final model is achieved. Data were also truncated more aggressively than is probably warranted.

However, the structures do appear adequate. The authors should really have attempted to address or correct the alerts that appeared in the models.

Body text corrections:

Abstract: Please rephrase the opening sentence. It is vague and confusing to the reader. Open with what the article discusses: the reactivity of potential heavy element ketone mimics.

pg 1, line 14: remove "Especially"

pg 1, line 31: np-orbitals --> p-orbitals

pg. 8 line 251: remove "in the unit cell"

pg. 8 line 247: Fe --> Se

pg. 10 line 284: argon ga, --> argon gas,

pg 10. line 316: ... was added reaction ... --> was added the reaction ...

pg. 11 line 342: ... gradually added in air triphenylphosphine ... --> gradually added, in air, triphenylphosphine ... 

(note: commas about "in air")

References:

Check formatting of references. There is a mix of capitalization and regular sentences for article names. Some consistency in the choice of formatting shows attention to detail.

pg. 14 line 492: subscript numbers in chemical formula

Check others throughout as well

pg. 14 line 508: MAIN GROUP CHEMISTRY is in all caps, should be in some more consistent form.

Comments on the Quality of English Language

The written language appears appropriate. The Abstract is the biggest issue with this submission.

Author Response

Thank you very much for the fruitful suggestions. Please see the attachment.
